# Gluconate-Lactobionate-Dextran Perfusion Solutions Attenuate Ischemic Injury and Improve Function in a Murine Cardiac Transplant Model

**DOI:** 10.3390/cells11101653

**Published:** 2022-05-16

**Authors:** Yinan Guo, Franka Messner, Sarah E. Beck, Marcos Iglesias Lozano, Hubert Schwelberger, Yichuan Zhang, Kai Kammers, Byoung Chol Oh, Elizabeth D. Greene, Gerald Brandacher, Kelvin G. M. Brockbank

**Affiliations:** 1Vascularized Composite Allotransplantation (VCA) Laboratory, Department of Plastic and Reconstructive Surgery, The Johns Hopkins University School of Medicine, Baltimore, MD 21205, USA; luciphilan@163.com (Y.G.); franka.messner@i-med.ac.at (F.M.); miglesi1@jhmi.edu (M.I.L.); yzhan317@jhu.edu (Y.Z.); boh3@jhmi.edu (B.C.O.); 2Department of Hand and Microsurgery, Xiangya Hospital, Central South University, Changsha 410017, China; 3Department of Visceral, Transplant and Thoracic Surgery, Medical University of Innsbruck, 6020 Innsbruck, Austria; hubert.schwelberger@i-med.ac.at; 4Department of Molecular and Comparative Pathobiology, The Johns Hopkins University School of Medicine, Baltimore, MD 21205, USA; sbeck8@jhmi.edu; 5Division of Biostatistics and Bioinformatics, Department of Oncology, Sidney Kimmel Comprehensive Cancer Center, The Johns Hopkins University School of Medicine, Baltimore, MD 21231, USA; kai.kammers@jhu.edu; 6Tissue Testing Technologies LLC, North Charleston, SC 29406, USA; bgreene236@gmail.com (E.D.G.); kgbrockbankassoc@aol.com (K.G.M.B.); 7Department of Bioengineering, Clemson University, Charleston, SC 29634, USA

**Keywords:** organ preservation, preservation solutions, heart transplant, static cold storage, murine model, animal research

## Abstract

Static cold storage is the cheapest and easiest method and current gold standard to store and preserve donor organs. This study aimed to compare the preservative capacity of gluconate-lactobionate-dextran (Unisol) solutions to histidine-tryptophan-ketoglutarate (HTK) solution. Murine syngeneic heterotopic heart transplantations (Balb/c-Balb/c) were carried out after 18 h of static cold storage. Cardiac grafts were either flushed and stored with Unisol-based solutions with high-(UHK) and low-potassium (ULK) ± glutathione, or HTK. Cardiac grafts were assessed for rebeating and functionality, histomorphologic alterations, and cytokine expression. Unisol-based solutions demonstrated a faster rebeating time (UHK 56 s, UHK + Glut 44 s, ULK 45 s, ULK + Glut 47 s) compared to HTK (119.5 s) along with a better contractility early after reperfusion and at the endpoint on POD 3. Ischemic injury led to a significantly increased leukocyte recruitment, with similar degrees of tissue damage and inflammatory infiltrate in all groups, yet the number of apoptotic cells tended to be lower in ULK compared to HTK. In UHK- and ULK-treated animals, a trend toward decreased expression of proinflammatory markers was seen when compared to HTK. Unisol-based solutions showed an improved preservative capacity compared with the gold standard HTK early after cardiac transplantation. Supplemented glutathione did not further improve tissue-protective properties.

## 1. Introduction

Organ transplantation has the potential to save the lives of millions of people suffering from end-stage organ failure [1]. By February 2020, 112,596 people were registered at United Network for Organ Sharing and waiting for a life-saving transplantation. However, in 2019, only 39,717 transplants were performed, demonstrating a significant discrepancy between organ supply and demand [2]. Yet, organ transplantation not only relies on available organ donors but, more importantly, also on well-orchestrated logistics due to the limited timespan organs can be stored outside of the human body [3]. Organ procurement and allocation are the driving forces behind transplantation, and in many instances, allocation is not successful in the short time frame of a few hours in which the organ needs to be transplanted [4]. To meet the demand for organs, donor recruitment, donor management, and organ allocation need to be refined and, even more importantly, novel strategies to ameliorate and extend organ preservation and storage need to be explored [5].

In addition to being the current gold standard, static cold storage (SCS) is the cheapest and easiest way to store and preserve donor organs [5]. Since the beginning of organ transplantation in the 1950s and 1960s, various perfusion solutions have been developed, with the goal of extending organ and tissue preservation during storage by supplying nutrients, electrolytes, and antioxidants [6,7,8,9]. Those multicomponent solutions are crucial for pH regulation as they can mitigate acidosis resulting from anaerobic glycolysis and subsequent lactate accumulation [10,11]. Most solutions employed for cells and tissues are ‘extracellular-type’ isotonic solutions (e.g., histidine-tryptophan-ketoglutarate (HTK), gluconate-lactobionate-dextran (Unisol)-low potassium (ULK)), with a plasma-like formulation of ions that mimics the normal extracellular environment of cells. In the presence of extracellular-type solutions, cells imbibe water and swell due to the oncotic pressure of the intracellular milieu [12]. Unisol-high-potassium (UHK), in contrast, is an ‘intracellular-type’ hypertonic solution (e.g., University of Wisconsin (UW) solution) that mimics high intracellular potassium and low sodium concentrations, which is designed to restrict the passive exchange of water and ions during hypothermic exposure when cell membrane pumps are inhibited [13].

This study aimed to assess the potency of four Unisol-based solutions in comparison to the well-established HTK solution in a murine model of heterotopic cardiac transplantation. Our expectation was that the high potassium version of Unisol (UHK) might be a better heart storage solution than ULK.

## 2. Materials and Methods

### 2.1. Experimental Animals

Eight- to twelve-week-old male Balb/c (H-2Kd; Jackson Laboratory, Bar Harbor, ME, USA) mice served as donors and recipients. All animals were housed under standard conditions with unrestricted access to water and food. All experiments were approved by the Animal Care and Use Committee of the Johns Hopkins University School of Medicine (#MO19M240).

### 2.2. Heterotopic Cardiac Transplantation

Balb/c mice hearts were transplanted into Balb/c recipients using a modified version of the technique previously described [14], outlined in brief below.

#### 2.2.1. Donor Procedure

The donor animals were anesthetized with 4% isoflurane inhalation (2% maintenance) and a midline incision was performed, exposing the inferior vena cava (IVC). After the injection of 200 IU of heparin, 3 mL of cardioplegic Euro Collins solution were flushed through the IVC.

The rib cage was subsequently opened and after the removal of the thymic tissue, the aorta and pulmonary vein were transected and the heart was flushed with 5 mL of cold (4 °C) perfusion solution. The IVC and superior vena cava and the pulmonary veins were ligated and transected. The heart was then stored in 5 mL of preservation solution for 18 h at 4 °C until reperfusion in the recipient.

#### 2.2.2. Recipient Procedure

The recipient animal was anesthetized with 4% isoflurane inhalation (2% maintenance). An incision was made from the jugular fossa to the mandibula, and the right external jugular vein and common carotid artery were dissected. After ligating and transecting both vessels, 2 cuffs were placed over the vessels (24 G arterial cuff; 22 G venous cuff) and secured with a 6-0 silk ligature after eversion of the vessel wall. For revascularization, first, the aorta was pulled over the arterial cuff and then the pulmonary artery was anastomosed with the external jugular vein using 6-0 silk. Finally, the vessel clamps were removed and the heart reperfused immediately.

### 2.3. Preservation Solutions and Experimental Groups

This study investigated the two Unisol-based formulations Unisol-high potassium (75 mM; “UHK”) and Unisol-low potassium (25 mM; ULK). The formulation of the Unisol solutions and primary modes of action are provided in Table 1.

Both formulations were further tested with and without the addition of glutathione (3 mM, 0.92 G/L; Glut). Unisol-based perfusates were compared to grafts flushed and stored in HTK solution. A detailed overview of the experimental groups is presented in Table 2.

### 2.4. Functional Assessment

In all animals, reperfusion (opening of vascular clamps until 7 min after reperfusion) was video recorded and retrospectively analyzed by two blinded and experienced microsurgeons. To assess functional recovery after reperfusion, the rebeating time (the time between reperfusion and the first sign of cardiac contraction (either of the ventricles or atriums)) was recorded. The macroscopic aspects of the transplanted hearts at 2, 5, and 7 after reperfusion and on postoperative day (POD) 3 are provided in the Appendix A.

In addition, cardiac function was assessed at 2, 5, and 7 min and on postoperative day 3 after transplantation using a well-established modified functional score (Table 3) [15,16].

### 2.5. Histopathology and Immunohistochemistry

Upon reaching the study endpoint on POD 3, cardiac tissue was collected, and animals were euthanized by CO_2_ inhalation. Samples were fixed in 10% neutral buffered formalin and dehydrated in graded ethanol. Fixed tissues were further embedded in paraffin, sectioned at 5 µm, and stained with hematoxylin and eosin (H&E). All slides were reviewed and scored by an expert veterinary pathologist in a blinded fashion. Ischemia/reperfusion injury was graded according to a published 6-tier scoring system (score 0: no inflammation; score 1: cardiac infiltration in up to 5% of the cardiac sections; score 2: 6% to 10%; score 3: 11% to 30%; score 4: 31% to 50%; and score 5: >50% cardiac infiltration) [17].

Additionally, terminal transferase-mediated dUTP nick-end labeling (TUNEL) of tissue sections was performed using the In Situ Cell Death Detection Kit POD (Roche, Vienna, Austria) according to the manufacturer’s instructions, followed by counterstaining with Mayer’s hemalum solution (Merck, Darmstadt, Germany). For each slide, the number of TUNEL-positive cell nuclei was determined in 3 non-overlapping high-power fields (HPF; 40× objective) in a blinded fashion. Images were taken with a Nikon Eclipse E60 (Nikon, Minato City, Tokyo, Japan) using a 10× and 20× objective, respectively.

### 2.6. Serum Collection and Enzyme-Linked Immunosorbent Assay

Whole blood was collected 24 h after reperfusion from the submandibular vein and spun at 5000 rpm for 10 min. Serum enzyme levels of myoglobin were assessed using the Mouse Myoglobin ELISA Kit (#MYO-1; Life Diagnostics Inc., West Chester, PA, USA). The immunoassay was performed according to the manufacturer’s instruction.

### 2.7. Quantitative Polymerase Chain Reaction

Frozen collected tissues were mechanically grinded, lysed in TRIzol reagent (Thermo Fisher Scientific Inc., Waltham, MA, USA), and RNA extracted using chloroform (Thermo Fisher Scientific Inc., Waltham, MA, USA) and the RNeasy MiniElute Cleanup kit (Qiagen, Hilden, Germany). In total, 2 µg of RNA were reverse transcribed for first-strand cDNA synthesis using the SuperScriptTM IV Reverse Transcriptase kit (Thermo Fisher Scientific Inc., Waltham, MA, USA), according to the manufacturer’s protocol. Real-time qPCR was performed on a QuantiStudio 12k Flex Real Time PCR system (Thermo Fisher Scientific Inc., Waltham, MA, USA) calibrated for SYBR Green detection (Thermo Fisher Scientific Inc., Waltham, MA, USA), and specific pairs of primers were used to measure the expression of the following genes: IL-1β (forward 5′-TGTGCAAGTGTCTGAAGCAGC-3′ and reverse 5′-TGGAAGCAGCCCTTCATCTT-3′); TNF-α (forward 5′-CTGTAGCCCACGTCGTAGC-3′ and reverse 5′-TTGAGATCCATGCCGTTG-3′); HO-1 (forward 5′-GCCGAGAATGCTGAGTTCATG-3′ and reverse 5′-TGGTACAAGGAAGCCATCACC-3′); Nrf2 (forward 5′-TTCTTTCAGCAGCATCCTCTCCAC-3′ and reverse 5′-ACAGCCTTCAATAGTCCCGTCCAG-3′); iNOS (forward 5′-GGCAGCCTGTGAGACCTTTG-3′ and reverse 5′-CATTGGAAGTGAAGCGTTTCG-3′); and 18s (forward 5′-CGCCGCTAGAGGTGAAATTCT-3′ and reverse 5′-CGAACCTCCGACTTTCGTTCT-3′) (all from SigmaMillipore, St. Louis, MO, USA). Results were normalized to 18 s expression and analyzed using the Delta-delta Ct method. The relative levels of each gene were then compared across multiple samples.

### 2.8. Statistical Analysis

Results are expressed as mean and standard deviation (SD) and median and interquartile range (IQR) or range. The Shapiro-Wilk test was used to assess normal distribution. Analysis of variance (ANOVA) with Tukey’s multiple comparisons test was performed in case of normal distribution. The Mann-Whitney test and Kruskal–Wallis and Friedman test corrected with Dunn’s multiple comparison test were used for non-normal distributed data. All tests were 2-sided and a *p*-value of <0.05 was considered statistically significant. Prism GraphPad 9 (GraphPad Inc., San Diego, CA, USA) was used for all statistical tests.

## 3. Results

### 3.1. Rebeating Time and Functional Recovery

In control animals that received cardiac grafts that were flushed and stored in HTK, the median rebeating time was 119.5 s (IQR: 94.5–173.5), which was significantly prolonged in comparison to grafts that were treated with the Unisol-based perfusates UHK (56 s [IQR: 40–81], *p* < 0.0001), UHK + Glut (44 s [IQR: 37.5–58], *p* < 0.0001), ULK (45 s [IQR: 43.5–57], *p* < 0.0001), and ULK + Glut (47 s [IQR: 35–57], *p* < 0.0001). No significant differences were observed in the rebeating times between the Unisol-based perfusates (Figure 1A and Table 4).

Functional recovery was assessed at two, five, and seven minutes after reperfusion and on POD three (Figure 1B and Table 5).

While HTK-treated animals (median functional score 1, IQR: 0–1) only showed cardiac fibrillations two minutes post-reperfusion, ULK (median 2, IQR: 1.25–2; *p* < 0.001) and ULK + Glut (median 2, IQR: 1–2; *p* = 0.014) already demonstrated weak or partial contractions. No statistically significant differences in functional recovery were observed for the other Unisol-based perfusates (UHK: median 1, IQR: 0.25–2, *p* = 0.310; UHK + Glut: median 1, IQR: 1–1, *p* = 0.440) at two minutes compared to HTK. Five minutes after reperfusion, all Unisol-based perfusates showed weak or partial contractions while HTK still mostly demonstrated cardiac fibrillations (median 1, IQR: 1–2). Only ULK remained significantly superior in functionality compared to HTK at five minutes post-reperfusion (median 2, IQR: 1.25–3, *p* = 0.015). At seven minutes, a comparable functional scoring was seen between the HTK (median 2, IQR: 1.25–2) and Unisol-based groups (UHK: median 2.5, IQR: 2–3, *p* = 0.590; ULK: median 2.5, IQR: 2–3, *p* = 0.450; UHK + Glut: median 2, IQR: 1–3, *p* = 0.990; ULK + Glut: median 1, IQR: 1–2.75, *p* = 0.990). On POD three, hearts that were treated with the Unisol-based perfusates showed significantly better functional scores (UHK: median 4, range: 3–4, *p* < 0.001; ULK: median 3, range: 3–4, *p* = 0.009; UHK + Glut: median 3.5, range: 3–4, *p* = 0.002; ULK + Glut: median 3, range: 2.25–4, *p* = 0.032) compared to HTK (median 1.5, range: 0–3) (Figure 1B and Table 5).

### 3.2. Serum Markers

Median myoglobin levels were 93.7 (IQR: 11.7–1045.0), 38.4 (IQR: 6.9–246.2), 74.9 (IQR: 10.9–235.9), 95.9 (IQR: 2.9–175.6), and 22.5 ng/mL (IQR: 8.1–66.1) for HTK, UHK, UHK + Glut, ULK, and ULK + Glut, respectively (Figure 2).

Animals treated with UHK and ULK + Glut both demonstrated the lowest myoglobin serum levels 24 h after reperfusion. Nevertheless, these differences did not reach statistical significance (*p* = 0.990).

### 3.3. Histomorphologic Assessment

Ischemia/reperfusion injury was semi-quantitatively assessed by a six-tier scoring system according to the severity of inflammatory infiltration (Figure 3A and Figure 4).

Compared to naïve tissue samples (median 0 [IQR: 0–0.5]), ischemic injury led to a significant increase in the infiltrating cells in cardiac tissue samples in HTK (median 4 [IQR: 3–5], *p* = 0.024), UHK + Glut (median 4 [IQR: 2–4], *p* = 0.015), and ULK + Glut (median 4 [IQR: 3–5], *p* = 0.036). In contrast, a comparable inflammation score was observed in hearts treated with UHK (median 3 [IQR: 3–3], *p* = 0.310) and ULK (median 3 [IQR: 1.5–4], *p* = 0.150) compared to naïve cardiac tissue. No statistical differences (*p* = 0.990) in the inflammatory scores were observed between the treatment groups.

Cell apoptosis, assessed by TUNEL staining (Figure 3B and Figure 5), revealed an increase in apoptotic cells in cardiac tissue subjected to ischemic injury (naïve: median 0 [IQR: 0–0.3] vs. HTK: median 3 [IQR: 3–4.3], *p* < 0.001; UHK: median 3 [IQR: 1.7–3.7], *p* = 0.013; UHK + Glut: median 2.7 [IQR: 1.7–3.7], *p* = 0.022; ULK: median 1.7 [IQR: 0.8–2.3], *p* > 0.990; ULK + Glut: median 3 [IQR: 2.7–3.3], *p* = 0.002). No significant differences in the numbers of apoptotic cells were seen between HTK and Unisol-based perfusates; only cardiac tissue treated with ULK tended to exhibit a lower number of apoptotic cells (*p* = 0.078).

### 3.4. Expression Analysis

To further assess ischemia-induced tissue injury, qPCR was performed (Figure 6).

Heart grafts subjected to 18 h of SCS demonstrated an increased expression of the proinflammatory marker IL-1β. Compared to naïve cardiac tissue, samples treated with UHK + Glut (*p* = 0.032) and ULK + Glut (*p* = 0.018) demonstrated significantly increased levels while hearts treated with UHK and ULK both tended to exhibit lower IL-1β levels. Upregulation of the antioxidative enzyme HO-1 was seen in ischemically injured animals, which reached statistical significance in ULK and ULK + Glut compared to naïve hearts. TNF-α expression was significantly lower in the UHK group compared to naïve cardiac tissue (*p* = 0.021). Nrf-2 expression was lowest in the samples of HTK-treated animals and highest in ULK + Glut. iNOS expression was significantly decreased in UHK (*p* = 0.018) compared to naïve cardiac tissue and a general trend towards decreased iNOS expression after ischemic injury was observed.

## 4. Discussion

Preservation solutions, regardless of whether they are designed to be intracellular- or extracellular-type solutions, contain various components that aim to ameliorate ischemia/reperfusion injury, including (1) buffer systems to prevent acidosis, (2) oncotic active agents that prevent intracellular water influx and thus cell edema and swelling, (3) high-energy compounds that provide substrates for regeneration during cold storage and upon reperfusion, and (4) antioxidants to mitigate oxidative stress [18,19,20]. Unisol contains HEPES, a zwitterion buffer effective for pH control in cold temperatures; large molecules lactobionic acid, gluconate, and dextran with oncotic activity; and glucose, sucrose, and adenosine as energy substrates (Table 1). It was developed as a base formulation for hypothermic exposure that could be customized by the addition of additives that optimize its preservative capacity for the diverse requirements of different cells and tissues [20,21,22]. HTK, in contrast, consists of a strong histidine buffer system, with mannitol as the osmotic active agent, and α-ketoglutarate as the energy substrate. HTK is commonly used for cardiac preservation and with its low potassium concentration demonstrated superior outcomes in several studies compared to other cardioplegic solutions [23].

In this study, the potency of Unisol-based perfusates UHK and ULK with and without supplemented glutathione was investigated in a syngeneic murine cardiac transplant model. A syngeneic model was used to exclude the bias of alloimmune activation. All Unisol-based perfusates, regardless of their potassium concentration, led, compared to HTK, to a significantly improved clinical outcome as evidenced by faster rebeating of the heart grafts after reperfusion and better contractility early after reperfusion and especially at the endpoint on POD three. Though a distinct increase in inflammatory infiltrates and the frequency of apoptotic cells was seen in ischemically injured cardiac grafts, UHK and ULK showed lower inflammation scores than their counterparts with supplemented glutathione and HTK. Ischemic injury further led to an increase in the number of apoptotic cells in all samples subjected to cold ischemia, and only muscle tissue stored in ULK tended to exhibit less apoptosis. In comparison to HTK, a trend toward decreased expression of proinflammatory markers IL-1β and TNF-α was seen in UHK-treated animals. While HO expression was increased after ischemic injury, the expression levels of its regulator molecule, Nrf-2, remained stable in all groups when compared to the baseline expression. iNOS expression was decreased in cardiac tissue subjected to 18 h of SCS and reached statistical significance for UHK compared to naïve cardiac tissue. Together, these results suggest that both base solutions had lower expression of proinflammatory marker expression than their glutathione-treated counterparts. Especially for UHK, the expression of proinflammatory markers tended to be lower than for HTK, whereas the expression of Nrf-2 was higher in all Unisol-based perfusates compared to HTK. Myoglobin is a cardiac protein that is essential for oxygen storage and contributes to nitric oxygen homeostasis under physiological and pathological conditions. Upon myocardial injury, it is systemically released as one of the earliest injury markers, and subsequently rapidly cleared, especially when successful revascularization is achieved [24,25,26]. Our study found similarly low levels of myoglobin 24 h after reperfusion, indicating that all investigated cardiac grafts were successfully reperfused. Due to the dynamics of myoglobin release, this rather late timepoint might not be able to adequately reflect the actual myocardial injury; however, earlier blood sampling resulted in a higher postoperative mortality.

Unisol has thus far been investigated for preservation of pancreatic beta-cells and in the setting of swine clinical suspended animation [18,20]. For beta-cells, Campbell et al. demonstrated superior cell viability, and lower levels of apoptotic activity in Unisol-stored cells compared to ones stored in UW and Belzer’s machine perfusion solution [18,19].

In contrast to several other studies, our data yielded no signs of improvement in cardiac function nor tissue preservation after supplementation with the antioxidant glutathione [27,28,29]. This is in line with results from a study on rat cardiac ischemia/reperfusion injury that demonstrated similar functional scores in cardiac grafts subjected to 10 h of cold storage with and without added glutathione in the HTK solution [30].

Based on preliminary experiments in a murine cervical heterotopic cardiac transplant model, when SCS times ranging from 12 to 24 h (data not shown) were investigated, 18 h of SCS was chosen as the preferred ischemic duration. This time led to significant ischemic injury while still resulting in the rebeating of transplanted grafts and low overall mortality. In addition, the heterotopic cervical cardiac transplant model was used based on the facts that it allowed for easier clinical monitoring than the heterotopic abdominal technique [14,31]. Postoperative day three was chosen as the endpoint for this study, as this is a favorable timepoint to assess differences in the response to ischemic injury in this murine cardiac transplant model [32,33].

Limitations of this study include the low but well-accepted number of animals based on a power calculation that could still lead to bias due to single deviating results. In addition, the heterotopic heart transplant model is not a true ‘working heart’ setting; thus, it requires further testing in such a setting to assess the true impact of preserving cardiac function. As the heterotopic localization of the heart prevents it from generating measurable cardiac output (the blood flow is retrograde), cardiac functional assessment was limited to measuring the rebeating time and use of a well-established functional score that has been frequently used in this model. Due to the study design with the endpoint at POD three, no conclusion regarding long-term outcomes and chronic vascular and tissue changes can be made. As this was the first assessment of Unisol-based perfusates in the setting of cardiac transplantation, this study was not designed to investigate underlying mechanisms of action. To address the long-term effects of Unisol-based perfusates, additional studies are warranted. Currently, longer static cold storage times in a heterotopic heart transplant model are being investigated, and preclinical small animal studies in the setting of vascularized tissue allotransplantation and kidney transplantation are planned.

## 5. Conclusions

In summary, our data demonstrates that all cardiac isografts stored in Unisol-based perfusates were superior in terms of clinical outcomes, such as rebeating and functionality, compared to hearts treated with HTK. The two solutions UHK and ULK showed, compared to the other tested perfusates, the lowest microscopic signs of inflammation and apoptosis. Supplemented glutathione did not further improve the histomorphology and functional recovery. Unisol-based solutions, especially UHK and ULK, showed an improved preservative capacity compared with the gold standard HTK early after cardiac transplantation. The presence of high potassium concentrations in Unisol-based solutions had no significant benefits compared to the low-potassium version.

## Figures and Tables

**Figure 1 cells-11-01653-f001:**
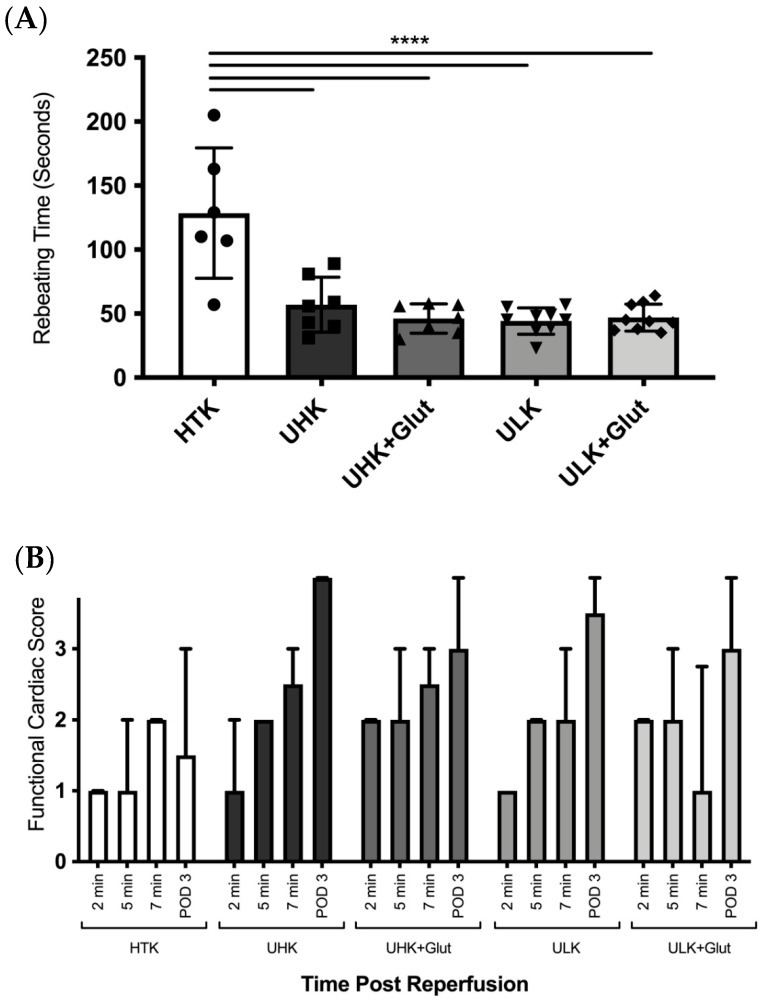
(**A**) Rebeating times, and (**B**) functional assessment after reperfusion. (**A**) Rebeating times were significantly (*p* < 0.0001; ANOVA with Dunnett’s multiple comparison correction) shorter in all investigated Unisol-based solutions compared to HTK. (**B**) Functional cardiac score at two, five, and seven minutes and on POD three after reperfusion. *Unisol, gluconate-lactobionate-dextran; HTK, histidine-tryptophan ketoglutarate solution; UHK, Unisol-high-potassium; ULK, Unisol-low-potassium; Glut, glutathione; min, minutes; POD, postoperative day.* **** *p* < 0.0001.

**Figure 2 cells-11-01653-f002:**
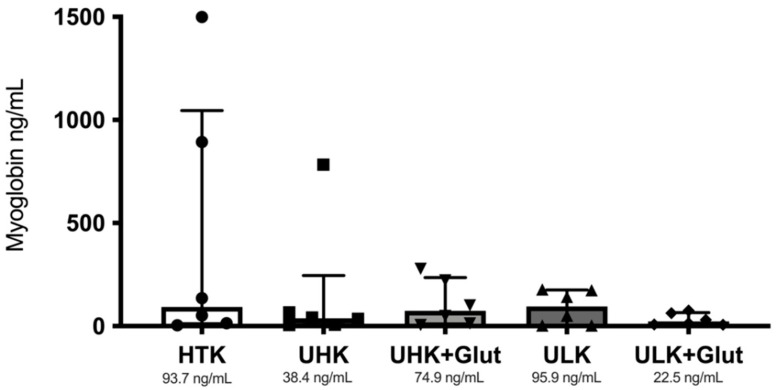
Myoglobin enzyme-linked immunosorbent assay (ELISA) for the quantification of myoglobin release 24 h after reperfusion. Similar (*p* = 0.900) serum myoglobin levels (median levels are shown below the graphs) were found for all investigated preservation solutions using the Kruskal-Wallis test. *Unisol, gluconate-lactobionate-dextran; HTK, histidine-tryptophan ketoglutarate solution; UHK, Unisol-high-potassium; ULK, Unisol-low-potassium; Glut, glutathione.*

**Figure 3 cells-11-01653-f003:**
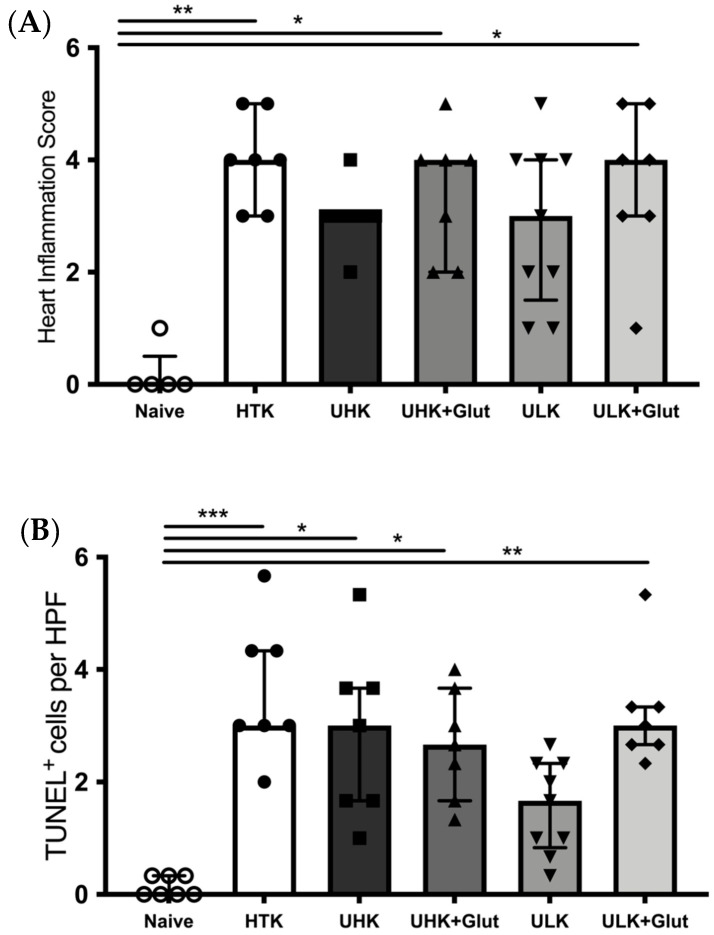
(**A**) Heart inflammation score and (**B**) assessment of apoptosis using TUNEL staining. (**A**) H&E-stained tissue sections were evaluated for the amount of cellular infiltrate using a semiquantitative 6-tier scoring system (score 0: no inflammation; score 1: cardiac infiltration in up to 5% of the cardiac sections; score 2: 6% to 10%; score 3: 11% to 30%; score 4: 31% to 50%; and score 5: >50% cardiac infiltration). All investigated solutions showed a comparable score that was significantly higher than in naïve hearts (n = 7–9/group). (**B**) Quantitative assessment of TUNEL^+^ cells (n = 7–9/group). The Kruskal-Wallis test and Dunn’s multiple comparison test were used for intergroup comparison. *Unisol, gluconate-lactobionate-dextran; HTK, histidine-tryptophan ketoglutarate solution; UHK, Unisol-high-potassium; ULK, Unisol-low-potassium; Glut, glutathione; HPF, high-power field.* * *p* < 0.05, ** *p* < 0.01, *** *p* < 0.001.

**Figure 4 cells-11-01653-f004:**
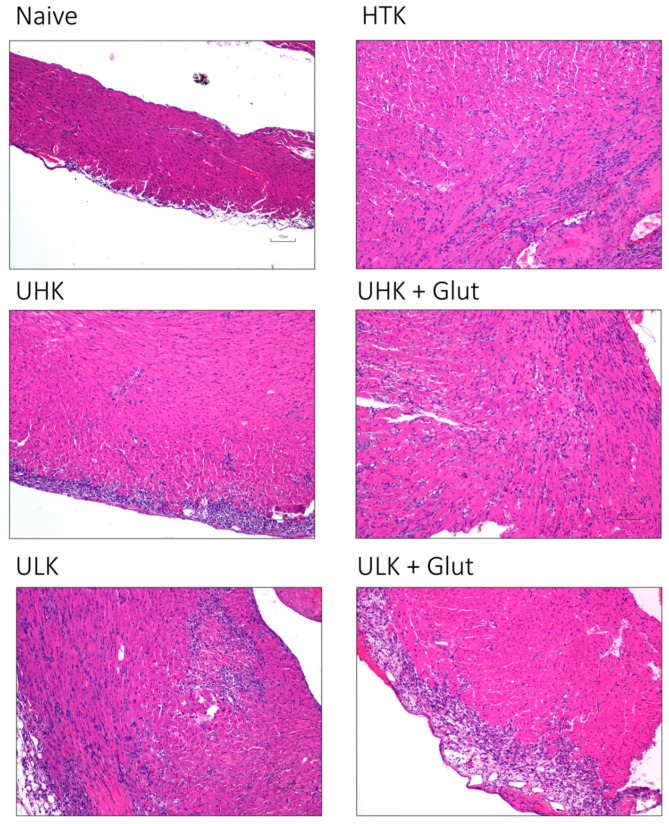
Histomorphologic assessment of cardiac tissue after heterotopic transplantation using H&E staining. Representative images of H&E-stained cardiac tissue sections taken on POD three after heterotopic heart transplantation (scale bar 100 µm, 10× magnification). *Unisol, gluconate-lactobionate-dextran; HTK, histidine-tryptophan ketoglutarate solution; UHK, Unisol-high-potassium; ULK, Unisol-low-potassium; Glut, glutathione.*

**Figure 5 cells-11-01653-f005:**
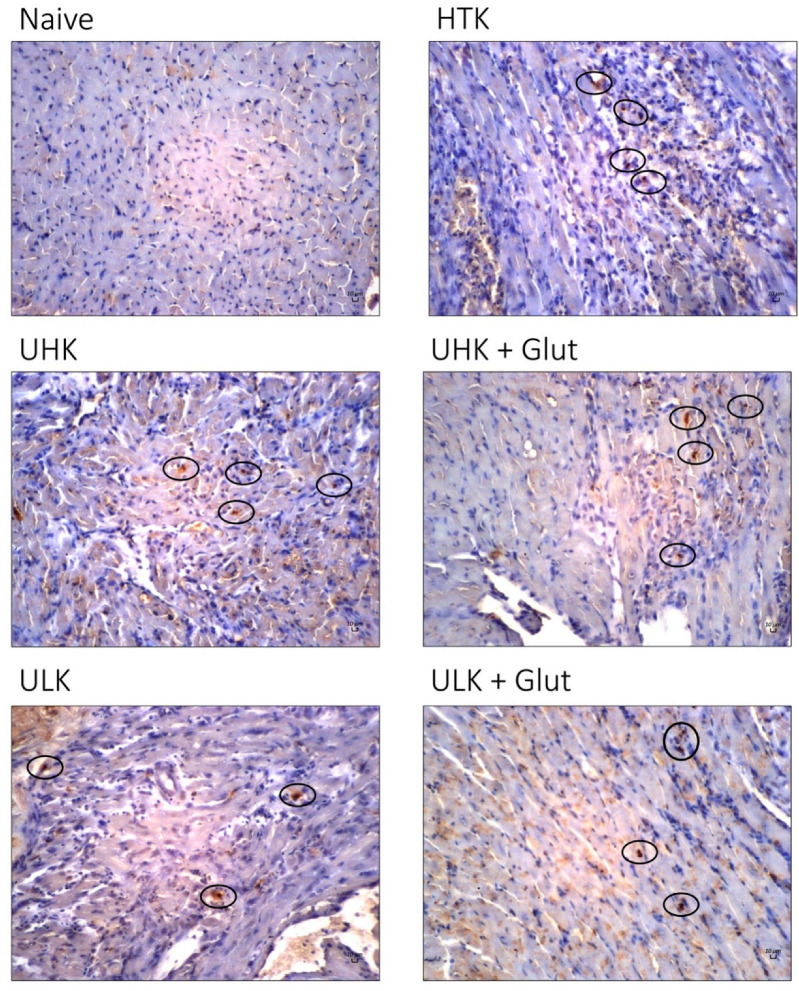
Representative images of TUNEL staining of cardiac tissue sections taken on POD three after heterotopic heart transplantation (scale bar 10 µm; 20× magnification). Compared to naïve cardiac tissue, all experimental groups displayed significantly increased numbers of TUNEL^+^ cells (dark cells marked with black circles) in graft biopsies. *Unisol, gluconate-lactobionate-dextran; HTK, histidine-tryptophan ketoglutarate solution; UHK, Unisol-high-potassium; ULK, Unisol-low-potassium; Glut, glutathione.*

**Figure 6 cells-11-01653-f006:**
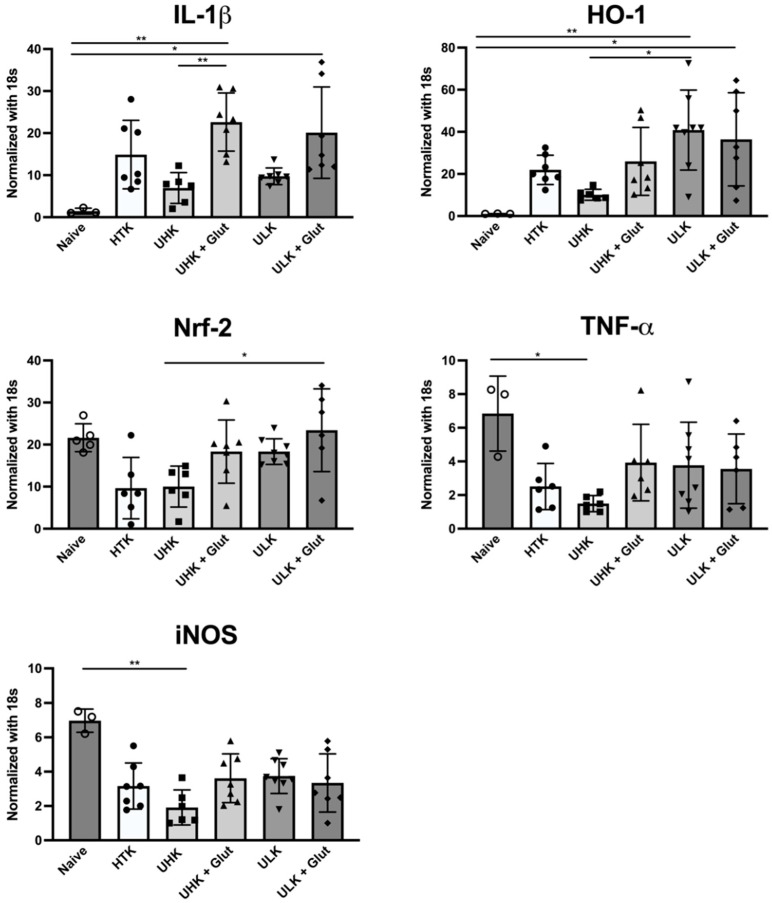
Expression analysis. Expression of pro- and anti-inflammatory markers IL-1β, TNF-α, HO-1, Nrf-2, and iNOS in cardiac tissue three days after heterotopic cardiac transplantation using qPCR. Cardiac tissues of naïve animals and heterotopically transplanted hearts subjected to 18 h of SCS that were treated with either HTK, UHK, UHK + Glut, ULK, or ULK + Glut were analyzed (naive control n = 3; others n = 6–8/group) using the Kruskal–Wallis test and Dunn’s multiple comparison test. *Unisol, gluconate-lactobionate-dextran; HTK, histidine-tryptophan ketoglutarate solution; UHK, Unisol-high-potassium; ULK, Unisol-low-potassium; Glut, glutathione.* * *p* < 0.05, ** *p* < 0.01.

**Table 1 cells-11-01653-t001:** Gluconate-lactobionate-dextran (Unisol) perfusion solutions: formulations and mechanisms of Action (350 mOsmoles and pH 7.6) in comparison to HTK.

Base Components	Mechanism of Action	Supplier	Cat #	UHK mmol/L	ULK mmol/L	Base Components	HTK mmol/L
K-Gluconate	Large ions that balance intracellular ions when ion pumps are off in cold. Prevents cell swelling.	Sigma	G-4500	50	25	Mannitol	30
Na-Gluconate		Sigma	G-9005	20	45		
Lactobionic acid		Sigma	L-2398	30	30		
NaOH (1N soln)	Ions and pH (final pH adjusted using HCl and NaOH).	Fisher	SS266	40	50	NaCl	15
KOH (1N soln)		Fisher	SP208	15	0	KCl	9
NaH_2_PO_4_		Sigma	S-0751	2.5	0		
KH_2_PO_4_		Sigma	P-5655	0	2.5		
NaHCO_3_		Fisher	BP328-1	0	5		
KHCO_3_		Sigma	P-4913	5	0		
MgCl_2_-_6_H_2_O		Sigma	M-9272	15	15	MgCl_2_-_6_H_2_O	4
CaCl_2_-_2_H_2_O		Sigma	C-3881	0.05	0.05	CaCl_2_-_2_H_2_O	0.015
HEPES	Zwitterion pH buffer effective in the cold, prevents acidosis.	Sigma	H-3375	35	35	Ketoglutarate	1
Glucose	Substrates for regeneration of high energy compounds in the cold and stimulation of recovery upon rewarming and reperfusion.	Sigma	G-5767	5	5	Tryptophan	2
Sucrose		Sigma	S-5016	15	15	Histidine	198
Mannitol		Sigma	M-1902	25	25		
Adenosine		Sigma	A-4036	2	2		
Glutathione	In supplemented groups.	Sigma	G-4251	3	3		
Dextran 40,000 or 70,000	Oncotic pressure, prevents cell swelling.	MP Biomedical		6%	6%		

**Table 2 cells-11-01653-t002:** Experimental groups.

Group No.	Strain Combination	Ischemic Time (h)	Perfusate	Number
1	Balb/c-Balb/c	18 h	HTK	7
2	Balb/c-Balb/c	18 h	UHK	7
3	Balb/c-Balb/c	18 h	UHK + Glut	7
4	Balb/c-Balb/c	18 h	ULK	9
5	Balb/c-Balb/c	18 h	ULK + Glut	9

h, hours; HTK, histidine-tryptophan ketoglutarate solution; Unisol, Gluconate-lactobionate-dextran solutions; UHK, Unisol-high-potassium; ULK, Unisol-low-potassium; Glut, Glutathione.

**Table 3 cells-11-01653-t003:** Functional cardiac scores. Assessed two, five, and seven minutes after reperfusion and on postoperative day three.

score 0	no beating
score 1	fibrillations, no real contractions
score 2	weak or partial contractions
score 3	homogenous contractions of both ventricles at reduced frequency and intensity
score 4	normal contraction intensity and frequency

**Table 4 cells-11-01653-t004:** Rebeating times (seconds) after graft flush and storage with HTK and Unisol-based perfusates.

	HTK	UHK	UHK + Glut	ULK	ULK + Glut
**Median**	119.5	56	44	45	47
**IQR**	94.5–173.5	40–81	37.5–58	43.5–57	35–57

HTK, histidine-tryptophan ketoglutarate solution; UHK, Unisol-high-potassium; ULK, Unisol-low-potassium; Glut, glutathione; IQR, interquartile range.

**Table 5 cells-11-01653-t005:** Functional cardiac scores (median [IQR]) and intergroup comparison between HTK (reference) and Unisol-based perfusates using the Kruskal–Wallis test with Dunn’s multiple comparison correction. IQR, interquartile range.

	HTK	UHK	UHK + Glut	ULK	ULK + Glut
** *2 min* **	1 [0–1]	1 [0.25–2]	1 [1–1]	2 [1.25–2]	2 [1–2]
*p-value*	Ref.	0.310	0.440	<0.001	0.014
** *5 min* **	1 [1–2]	2 [2–2]	2 [1.25–2]	2 [1.25–3]	2 [1–3]
*p-value*	Ref.	0.054	0.360	0.015	0.072
** *7 min* **	2 [1.25–2]	2.5 [2–3]	2 [1–3]	2.5 [2–3]	1 [1–2.75]
*p-value*	Ref.	0.590	0.990	0.450	0.990
** *POD 3* **	1.5 [0–3]	4 [3–4]	3.5 [3–4]	3 [3–4]	3 [2–4]
*p-value*	Ref.	<0.001	0.002	0.009	0.032

Unisol, gluconate-lactobionate-dextran; HTK, histidine-tryptophan ketoglutarate solution; UHK, Unisol-high-potassium; ULK, Unisol-low-potassium; Glut, glutathione; min, minutes; POD, postoperative day.

## Data Availability

The data presented in this study are available on request from the corresponding author.

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
