# Peer review of "Gluconate-Lactobionate-Dextran Perfusion Solutions Attenuate Ischemic Injury and Improve Function in a Murine Cardiac Transplant Model"

_cells, 2022, doi:10.3390/cells11101653_

Round 1
Reviewer 1 Report
It is with great interest that I read the report on the use of UNISOL versus HTK as storage solution to attenuate ischemic injury and improve short term function in a murine cervical heart transplant model. The authors have concisely investigated the difference between these two solutions, with and without added Glutathione and qualified and quantified the short term results through visual inspection, histology and qPCR of pro-inflammatory markers. The results are clearly presented and easily graspable for a reader.
In the discussion, it could be clarified why the authors chose a three day follow-up as opposed to a more long term follow up. Furthermore, it would be of interest to mention future directions for further investigation of these storage solution for cardiac grafts as it seems to be performing better than the current gold standard.
Author Response
We thank the reviewer for the positive assessment of our manuscript. We have added information regarding the endpoint on POD three on page 14 Line 385-386 and described future projects on page 14 (lines 386-390).
Reviewer 2 Report
This manuscript addressed organ transplantation which is of great importance for potential clinical application. The authors present the article in an organized way and in good English.
- Please check all the Arabic numbers throughout the article, leave a space between the numbers and their units.
- Replace (a), (b), (c), (d) with (A), (B), (C) and (D) in Fig 1.
- Check all the molecular formula throughout the paper, especially in Table 1, make them correct.
- Check all the statistic “P”, put them as Capital Italics.
- There are three layers for the heart wall, endocardium, myocardium and epicardium. I'm wondering in which layer the increased leukocyte recruitment induced by ischemic injury mainly occur?
- Figure 2 needs to be improved by adding A, B, C, D on each panel.
Author Response
- We corrected the layout accordingly and thank the reviewer for paying attention to such details.
- We replaced them with capital letters.
- We checked all the molecular formulas and corrected those who were not adequately displayed.
- We checked the manuscript and put all P-values in capital italic letters.
- The tissue just beneath the inner layer of the heart, the subendocardial layer is most susceptible to ischemic injury and thus is the first to be infiltrated by leukocytes. In case of a more pronounced ischemic injury, it can also affect (depending on the severity) up to the whole thickness of the myocardium.
- We changed Figure 2 according to reviewers’ comments. In order to increase the quality, we split the former figure 2 in three separate figures. We paid special attention to labeling in the revised version.
Reviewer 3 Report
Dear Authors, thank you for opportunity to review your interesting work. I found this work methodical and reproducible, moreover presented results demonstrate improved preservative potential of UNISOL based solutions in contrast to standard used HTK. I think there is potential in follow up study with longer observation period.
I have some minor notes.
Methods 2.6 Serum collection… Please specify Myoglobin kit used.
Methods 2.7. qPCR… Please list probes used and their source.
Presenting numeric results – in subchapters 3.1., 3.2, 3.3
As you are describing numeric results in text, like line 201 “(UHK: median 2.5, IQR 2-3, P = 0.59; ULK: median 2.5, IQR 2-3, P = 0.45; 201 UHK+Glut: median 2, IQR 1-3, P = 0.99; ULK+Glut: median 1, IQR: 1-2.75, P = 0.99 )” I think the one table containing consolidating these numeric results will make easier to read and compare.
I would appreciate pictures of donor heart after cold perfusion, after suturing, declamping and after longer reperfusion – each group. These can be added into supplementary section.
Figure 1 – just cosmetic note about table, please use same precision of numbers
Figure 2 – I would suggest splitting histology images and charts in separate figures. It can be caused by low res in review pdf but in this form, it is complicated to observe HE figures. Please also add scale bar to at least one image (group). I don’t understand the different color between left and right group of HE images – the pictures were taken from different types of tissue or using different microscopes/scanners or different light conditions, white balance? Please also add information about microscope/scanner used and objective. There is some pink pale circle (sweet spot) in the middle of HE images (right group), is this caused by technique of staining; different type or composition of tissue; or by microscope light path generating this artifact?
Figure 3 - for better clarity, I would also add captions under bars in charts.
Author Response
We thank the reviewer for the constructive comments. We specified the myoglobin kit used for ELISA (2.6 Methods, page 5, lines 150-151). We added information on the probes and their source for qPCR (2.7 qPCR, page 5, lines 162-170). We included two tables (table 4 and table 5; page 6 and 7, respectively) that contain the numeric results presented in the results subchapter 3.1 and paid special attention to using the same precision of numbers in the figure and text. We added median serum myoglobin levels in Figure 2. We added representative images (n=5/group) of murine heart grafts at two-, five-, and seven minutes after reperfusion as well as on POD three (assessment timepoints) as supplementary section. Unfortunately, we did not take pictures of hearts after cold perfusion and after suturing, therefore we are unable to provide these images. We modified, in concordance with the reviewers’ suggestion, Figure 2 and separated histology images from graph charts. Figure 4 demonstrates representative H&E images of murine cardiac grafts on POD 3 (scale bar 100µm in right lower corner; due to overlap it is not equivalently well visible in all pictures) and Figure 5 shows pictures taken after TUNEL staining and counterstaining with Mayer's hemalum solution (scale bar 10 µm in right lower corner; due to overlap it is not equivalently well visible in all pictures). The differences in color are due to the fact that two different stainings were used. Due to missing labeling in our initial submission, this might have not been clear to the reader. Nevertheless, by separating both panels, we hope that this is clear now. In addition, we added the requested information on camera/objective (Methods 2.5; page 4, lines 146-147). We revised figure 3 and added captions under the bars in all charts to increase readability.